# Analytical Performance and Potential Clinical Utility of EUCAST Rapid Antimicrobial Susceptibility Testing in Blood Cultures after Four Hours of Incubation

Anna Ekwall-Larson,[a,b] Inga Fröding,[b,c] Berivan Mert,[a] Anna Åkerlund,[d,e] Volkan Özenci[a,b]

[a]Department of Clinical Microbiology, Karolinska University Hospital, Huddinge, Stockholm, Sweden
[b]Department of Laboratory Medicine, Division of Clinical Microbiology, Karolinska Institutet, Stockholm, Sweden
[c]Department of Microbiology, Public Health Agency of Sweden, Solna, Sweden
[d]Division of Clinical Microbiology, Department of Clinical and Experimental Medicine, Linköping University Hospital, Linköping, Sweden
[e]Division of Clinical Microbiology, Linköping University Hospital, Linköping, Sweden

**ABSTRACT** EUCAST rapid antimicrobial susceptibility testing (RAST) provides antibiotic susceptibility results after 4 to 8 h of incubation. This study assessed the diagnostic performance and clinical usefulness of EUCAST RAST after 4 h. This was a retrospective clinical study performed on blood cultures with *Escherichia coli* and *Klebsiella pneumoniae* complex (*K. pneumoniae* and *Klebsiella variicola*) at Karolinska University Laboratory (Stockholm, Sweden). The rate of categorized RAST results and the categorical agreement (CA) of RAST with the standard EUCAST 16-to-20-h disk diffusion (DD) method for piperacillin-tazobactam, cefotaxime, ceftazidime, meropenem, and ciprofloxacin were analyzed, as well as the utility of RAST for adjusting the empirical antibiotic therapy (EAT) and the combination of RAST with a lateral flow assay (LFA) for extended-spectrum β-lactamase (ESBL) detection. A total of 530 *E. coli* and 112 *K. pneumoniae* complex strains were analyzed, generating 2,641 and 558 readable RAST zones, respectively. RAST results categorized according to antimicrobial sensitivity/resistance (S/R) were obtained for 83.1% (2,194/2,641) and 87.5% (488/558) of *E. coli* and *K. pneumoniae* complex strains, respectively. The RAST result categorization to S/R for piperacillin-tazobactam was poor (37.2% for *E. coli* and 66.1% for *K. pneumoniae* complex). CA with the standard DD method was over 97% for all tested antibiotics. Using RAST, we detected 15/26 and 1/10 of the *E. coli* and *K. pneumoniae* complex strains that were resistant to the EAT. For patients treated with cefotaxime, RAST was used to detect 13/14 cefotaxime-resistant *E. coli* strains and 1/1 cefotaxime-resistant *K. pneumoniae* complex strain. ESBL positivity was reported the same day as blood culture positivity with RAST and LFA. EUCAST RAST provides accurate and clinically relevant susceptibility results after 4 h of incubation and can accelerate the assessment of resistance patterns.

**IMPORTANCE** Early effective antimicrobial treatment has been shown to be crucial for improving the outcome of bloodstream infections (BSI) and sepsis. In combination with the rise of antibiotic resistance, this calls for accelerated methods for antibiotic susceptibility testing (AST) for effective treatment of BSI. This study assesses EUCAST RAST, an AST method that yields results in 4, 6, or 8 h after blood culture positivity. We analyzed a high number of clinical samples of *Escherichia coli* and *Klebsiella pneumoniae* complex strains and confirm that the method delivers reliable results after 4 h of incubation for the relevant antibiotics for treating *E. coli* and *K. pneumoniae* complex bacteremia. Furthermore, we conclude that it is an important tool for antibiotic treatment decision-making and early detection of ESBL-producing isolates.

**KEYWORDS** ESBL, antibiotic resistance, blood culture, bloodstream infection, rapid tests, susceptibility testing

Address correspondence to Volkan Özenci, volkan.ozenci@ki.se.

The authors declare no conflict of interest.

Sepsis is defined as organ dysfunction due to a dysregulated response to infection and is associated with both high morbidity and mortality (1). Bloodstream infection (BSI) is the most common cause of sepsis. The outcome is strongly associated with early initiation of an active antibiotic agent (2–4). Treatment is commenced with an empirical broad-spectrum antibiotic and deescalated when the identity and susceptibility profile of the pathogen are known (5). The rising prevalence of antibiotic resistance enhances the risk of empirical antibiotic treatment (EAT) failure, thus increasing the need for rapid antibiotic susceptibility testing (AST) methods. Despite the development of rapid molecular AST methods, blood culture (BC)-based phenotypic AST methods remain the gold standard (6, 7). However, current standard AST methods have a turnaround time of 16 to 24 h after BC positivity.

To address this issue, the European Committee on Antimicrobial Susceptibility Testing (EUCAST) has developed a method for rapid antimicrobial susceptibility testing (RAST) directly from positive BC bottles (8, 9). EUCAST RAST is based on the EUCAST standard disk diffusion (DD) method and allows AST results in 4 to 8 h after BC positivity, using time- and species-specific DD breakpoints and introducing an area of technical uncertainty (ATU) to minimize errors (10). By combining susceptibility testing directly on positive BC broth with a shortened incubation time before zone reading, the turnaround time is significantly reduced compared to standard AST methods. A further advantage of the method is its similarity to the standard DD method, which facilitates implementation.

It is important to evaluate the effect of implementation of rapid methods in the clinical routine. However, hitherto published studies have provided limited information about the performance in relation to effectiveness and clinical usefulness, although showing reliable AST results with EUCAST RAST (11–17), including earlier initiation of optimal antibiotic therapy (12, 14). We aimed to evaluate the performance of EUCAST RAST for *E. coli* and *K. pneumoniae* isolates (i.e., *K. pneumoniae* complex, including *K. pneumoniae* and *Klebsiella variicola* isolates) compared to the standard DD method and the possible clinical impact of rapid susceptibility reports. The latter was evaluated by assessing the appropriateness of the EAT based on the EUCAST RAST results. The outcomes were the proportion of RAST results categorized as susceptible or resistant, categorical agreement with the reference method, and possible correction of ineffective EAT. To evaluate the potential clinical usefulness of RAST in antibiotic treatment decision-making, the cefotaxime RAST results for patients treated with piperacillin-tazobactam, meropenem, and ciprofloxacin were analyzed. Lastly, we wanted to assess the effect on the time to detection of extended-spectrum $\beta$-lactamase (ESBL)-producing *E. coli* and *K. pneumoniae* when combining EUCAST RAST with a lateral flow immune assay (LFA) (NG-Test CTX-M/CTX-M Multi; NG Biotech, Guipry, France). The outcome was the time to report of ESBL positivity in the laboratory information system (LIS).

## RESULTS

Blood cultures with Gram-negative bacteria that signaled positive before 11:00 a.m. were included in the study. In total, 530 *E. coli* and 112 *K. pneumoniae* complex isolates (97 *K. pneumoniae* isolates and 15 *K. variicola* isolates, from here on referred to as *K. pneumoniae* only) were included for analysis (Fig. 1), generating 2,650 and 559 standard DD AST results. The mean age of the patient population infected with *E. coli* was 71 ± 17 years, and 50.6% were male. For the *K. pneumoniae* patient group, the mean age was 69 ± 17 years, and 75.9% were male.

**RAST performance after 4 h of incubation.** In total, there was a readable RAST result for 99.7% (2,641/2,650) of the *E. coli* antibiotic tests and 99.8% (558/559) of the *K. pneumoniae* antibiotic tests. For the *E. coli* tests, 83.1% (2,194/2,641) of the results were categorized as susceptible (S) or resistant (R). The corresponding percentage for the *K. pneumoniae* test results was 87.5% (488/558). The highest rates of results in the ATU were observed with piperacillin-tazobactam for both *E. coli* and *K. pneumoniae* isolates (62.8% and 33.9%, respectively), followed by ciprofloxacin (10.6% and 20.9%, respectively). The proportion of results categorized as either S or R was >90% for cefotaxime, ceftazidime, and meropenem (Table 1). The overall categorical agreement (CA) was 98.6% (2,163/2,194) for the *E. coli* strains and 99.0% (483/488) for the *K. pneumoniae* strains. Overall, the percentages of very major

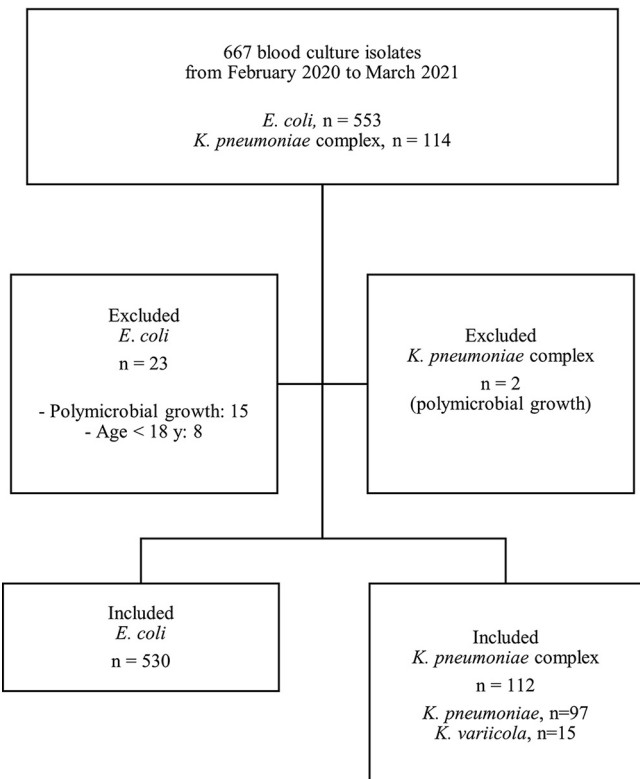

**FIG 1** Flow chart of the main study population.

errors (VMEs), major errors (MEs), and minor errors (mEs) were 0.4% (8/2,194), 0.6% (14/2,194) and 0.4% (9/2,194) for the *E. coli* strains and 1.0% (5/488), 0% (0/488), and 0% (0/488) for the *K. pneumoniae* strains, respectively, with piperacillin-tazobactam presenting with the highest error rate for both bacteria. For one *K. pneumoniae* isolate, the RAST result for meropenem was S and the standard DD result was R, i.e., a VME. This isolate was further analyzed according to the standard routine in the laboratory by broth microdilution (BMD) and categorized as susceptible with high increased exposure (I) according to the obtained MIC for meropenem (8 mg/L). Consequently, this RAST result was interpreted as an mE. The number of errors per antibiotic are presented in Table 1.

**RAST results in relation to empirical antibiotic treatment.** In patients with *E. coli* bacteremia, cefotaxime was the most common EAT (43.4%), while patients with *K. pneumoniae*-positive BCs were most often treated with piperacillin-tazobactam (32.1%) (Table 2). Patients with *E. coli* (*n* = 437) and *K. pneumoniae* (*n* = 87) bacteremia, treated with either piperacillin-tazobactam, cefotaxime, meropenem, or ciprofloxacin, were assessed for appropriateness of the EAT based on the RAST results. For these patients, 5.9% (26/437) of the *E. coli* and 11.5% (10/87) of the *K. pneumoniae* strains were resistant to the EAT, of which RAST detected 57.7% (15/26) and 10.0% (1/10), respectively (Table 3). For patients treated with cefotaxime, RAST detected nearly all EAT-resistant isolates (13/14 *E. coli* strains and 1/1 *K. pneumoniae* strain). However, for patients treated with piperacillin-tazobactam, none of the EAT-resistant isolates were detected by RAST (Table 3). For *E. coli* and piperacillin-tazobactam, the undetected resistant isolates were in the ATU. The cefotaxime-resistant *E. coli* isolate that was undetected by RAST was incorrectly categorized as S (i.e., VME).

**Determination of cefotaxime susceptibility and resistance.** For patients treated with piperacillin-tazobactam, meropenem, and ciprofloxacin, RAST detected 96.8% (30/31) and 100% (10/10) of the cefotaxime-resistant *E. coli* and *K. pneumoniae* isolates, respectively (Table 4). A correct cefotaxime S classification was obtained for 126/129 *E. coli* bacteremia cases treated with piperacillin-tazobactam, 24/26 *E. coli* bacteremia cases treated with meropenem, and 19/20 *E. coli* bacteremia cases treated with ciprofloxacin. For two patients with *E. coli* bacteremia and treatment with piperacillin-tazobactam as the EAT, RAST produced

**TABLE 1** Susceptibility testing results according to the reference method and RAST results for *E. coli* and *K. pneumoniae* complex strains[a]

| Microorganism | Outcome | Antibiotic agent, No. (%) | | | | |
| --- | --- | --- | --- | --- | --- | --- |
| | | PTZ | CTX | CAZ | MER | CIP |
| *E. coli* | DD method result | | | | | |
| | S | 483 (91.1) | 472 (89.1) | 472 (89.1) | 530 (100.0) | 419 (79.1) |
| | I | 9 (1.7) | 5 (0.9) | 4 (0.8) | 0 | 20 (3.8) |
| | R | 38 (7.2) | 53 (10.0) | 54 (10.2) | 0 | 91 (17.2) |
| | Readable RAST[b] | 529 (99.8) | 529 (99.8) | 527 (99.4) | 529 (99.8) | 527 (99.4) |
| | RAST S/R[c] | 197 (37.2) | 516 (97.5) | 488 (92.6) | 522 (98.7) | 471 (89.4) |
| | RAST ATU[c] | 332 (62.8) | 13 (2.5) | 39 (7.4) | 7 (1.3) | 56 (10.6) |
| | CA[d] | 191 (97.0) | 508 (98.4) | 482 (98.8) | 521 (99.8) | 461 (97.9) |
| | VME[d] | 0 | 2 (0.4) | 2 (0.4) | 0 | 4 (0.8) |
| | ME[d] | 5 (2.5) | 4 (0.8) | 3 (0.6) | 1 (0.2) | 1 (0.2) |
| | mE[d] | 1 (0.5) | 2 (0.4) | 1 (0.2) | 0 | 5 (1.1) |
| *K. pneumoniae* complex | DD method result | | | | | |
| | S | 83 (74.1) | 97 (86.6) | 93 (83.0) | 111 (99.1) | 82 (73.9) |
| | I | 2 (1.8) | 1 (0.9) | 1 (0.9) | 0 | 8 (7.2) |
| | R | 27 (24.1) | 14 (12.5) | 18 (16.1) | 1 (0.9)[e] | 21 (18.9) |
| | Readable RAST | 112 (100.0) | 112 (100.0) | 112 (100.0) | 112 (100) | 110 (99.1) |
| | RAST S/R | 74 (66.1) | 109 (97.3) | 107 (95.5) | 111 (99.1) | 87 (79.1) |
| | RAST ATU | 38 (33.9) | 3 (2.7) | 5 (4.5) | 1 (0.9) | 23 (20.9) |
| | CA | 72 (97.3) | 109 (100.0) | 106 (99.1) | 110 (99.1) | 86 (98.9) |
| | VME | 2 (2.7) | 0 | 1 (0.9) | 1 (0.9)[e] | 1 (1.1) |
| | ME | 0 | 0 | 0 | 0 | 0 |
| | mE | 0 | 0 | 0 | 0 | 0 |

[a]*E. coli* strains, *n* = 530; *K. pneumoniae* complex strains, *n* = 112 (*n* = 111 for ciprofloxacin because of one missing standard disk diffusion result). ATU, area of technical uncertainty; CA, categorical agreement; VME, very major error; ME, major error; mE, minor error; PTZ, piperacillin-tazobactam; CTX, cefotaxime; CAZ, ceftazidime; MER, meropenem; CIP, ciprofloxacin.
[b]Readable RAST results (proportion of all analyzed isolates).
[c]RAST-categorized S/R and ATU results (proportion of readable RAST results).
[d]Categorical agreement and errors (proportion of categorized RAST results).
[e]This isolate was categorized as R by standard disk diffusion but was further analyzed with broth microdilution (BMD) assay and categorized as I (i.e., mE according to the BMD result).

one VME and one ME for cefotaxime. The corresponding numbers for the *K. pneumoniae* bacteremia cases treated with piperacillin-tazobactam, meropenem, and ciprofloxacin were 31/31, 8/9, and 4/4, respectively, with no categorical errors.

**Time to ESBL report.** Unique and complete RAST results for 1,223 *E. coli* isolates and 268 *K. pneumoniae* isolates (all of these isolates were *K. pneumoniae* and none of them were *K. variicola*) were evaluated for cefotaxime or ceftazidime resistance. From this cohort, 137 ESBL-positive isolates (114 *E. coli* isolates and 23 *K. pneumoniae* isolates) were included for analysis of the time to ESBL report. For 75.9% (104/137) of these isolates, a positive LFA result was reported in the LIS. The median (interquartile range [IQR]) time to ESBL report in the LIS was 4.0 h (0.6 h) for RAST with a positive, reported LFA result, compared to 23.5 h (3.7 h) for the isolates in the RAST group without a reported LFA result ($P < 0.0001$) (Fig. 2).

## DISCUSSION

The rise of antibiotic resistance and the consequent risk of ineffective EAT demands rapid AST methods. The present study compared the performance of EUCAST RAST with the EUCAST standard disk diffusion method for *E. coli* and *K. pneumoniae* isolates after 4 h of incubation. Furthermore, we assessed the possible usefulness of EUCAST RAST for signaling ineffective EAT and the use of EUCAST RAST in combination with an LFA for a prompt ESBL report. The proportion of results that could be categorized as S or R was between 79.1% and 99.1% for cefotaxime, ceftazidime, meropenem, and ciprofloxacin, and the error rates were low, with CA values between 97.9% and 100%. However, the benefits of early reading were limited for piperacillin-tazobactam, for which only 37.2% of the *E. coli* isolates and 66.1% of the *K. pneumoniae* isolates could be categorized. EUCAST RAST provided relevant resistance information in relation to the EAT, especially for cefotaxime. Lastly, in combination with an LFA, the time to report of a positive ESBL result in the LIS was shortened by 19.5 h and was reported within a workday of BC positivity.

**TABLE 2** Empiric antibiotic treatment by number of patients[a]

| Microorganism | Antibiotic treatment | No. (%) of patients |
|---|---|---|
| E. coli | Cefotaxime | 230 (43.4) |
| | Piperacillin-tazobactam | 146 (27.5) |
| | Meropenem | 39 (7.4) |
| | No information available | 35 (6.6) |
| | No antibiotic treatment | 24 (4.5) |
| | Ciprofloxacin | 22 (4.2) |
| | Combination therapy[b] | 13 (2.5) |
| | Other Gram-negative treatment[c] | 9 (1.7) |
| | Gram-positive treatment[d] | 7 (1.3) |
| | Deceased (not evaluable) | 5 (0.9) |
| K. pneumoniae complex | Piperacillin-tazobactam | 36 (32.1) |
| | Cefotaxime | 32 (28.6) |
| | Meropenem | 15 (13.4) |
| | No information available | 8 (7.1) |
| | No antibiotic treatment | 6 (5.4) |
| | Gram-positive treatment[e] | 5 (4.5) |
| | Other Gram-negative treatment[f] | 4 (3.6) |
| | Ciprofloxacin | 4 (3.6) |
| | Combination therapy[g] | 2 (1.8) |

[a]Total n = 530 patients with E. coli bacteremia; n = 112 patients with K. pneumoniae complex bacteremia.
[b]Combined Gram-negative treatment: cefotaxime + gentamicin, cefotaxime + ciprofloxacin, cefotaxime + trimethoprim-sulfamethoxazole, meropenem + ciprofloxacin, meropenem + ciprofloxacin + colistin, meropenem + levofloxacin, piperacillin-tazobactam + gentamicin, piperacillin-tazobactam + trimethoprim-sulfamethoxazole, meropenem + trimethoprim-sulfamethoxazole.
[c]Benzylpenicillin + gentamicin, ertapenem, imipenem, metronidazole + trimethoprim-sulfamethoxazole.
[d]Amoxicillin, amoxicillin-clavulanic acid, cloxacillin, benzylpenicillin, phenoxymethylpenicillin.
[e]Cloxacillin, flucloxacillin, moxifloxacin, benzylpenicillin.
[f]Ceftazidime + metronidazole + vancomycin, ceftibuten, ceftolozane-tazobactam, trimethoprim-sulfamethoxazole.
[g]Combined Gram-negative treatment: ceftazidime + ciprofloxacin, piperacillin-tazobactam + trimethoprim-sulfamethoxazole.

In the present study, the highest ATU rates were due to piperacillin-tazobactam and ciprofloxacin, which is similar to previous studies (11–15, 18, 19). Nevertheless, the CA of EUCAST RAST with the standard DD method was well over 90% for all tested antibiotics. The highest VME rate of 2.7% (2/74) was for piperacillin-tazobactam and K. pneumoniae. Previous studies have also reported high rates of errors for piperacillin-tazobactam (13, 18, 19), except for Berinson et al., who reported cefotaxime as the cause of most VMEs (12). In the present study, cefotaxime was associated with low error rates (0.4% VMEs, 0.8% MEs, and 0.4% mEs for E. coli isolates and none for K. pneumoniae isolates). Both E. coli isolates that were incorrectly classified as S for cefotaxime by RAST were from BCs containing two different strains of E. coli (one cefotaxime-susceptible and one cefotaxime-resistant), which complicated the analysis. It is reasonable to suggest that the resistant strains could not be detected by EUCAST RAST due to their lower concentrations in the BC broth and consequently on the EUCAST RAST plates, compared to the susceptible strains. This interpretation contrasts with the two VMEs for piperacillin-tazobactam and K. pneumoniae, which were both from BCs with only one strain. Of note is that the RAST zones for both these isolates were 15 mm, which is the S breakpoint for piperacillin-tazobactam, and thus they were close to the ATU. In contrast to the results of Berinson et al., our results suggest that EUCAST RAST is efficient and reliable for detection of cefotaxime resistance. This finding is in line with a previous study in our laboratory on the short-term incubation of EUCAST disk diffusion, ROSCO ESBL, and carbapenemase detection kits in which we reported high performance for detection of cefotaxime resistance (20) and with a study by Cortazzo et al., who reported a CA of 100% for cefotaxime and E. coli and K. pneumoniae strains after 4, 6, and 8 h of incubation, including detection of all 48 ESBL-producing isolates (11).

Using EUCAST RAST, we detected 57.7% of the E. coli and 10.0% of the K. pneumoniae isolates that were resistant to the prescribed EAT and almost all of the isolates that were resistant to cefotaxime. However, EUCAST RAST did not detect any of the empirically

**TABLE 3** Susceptibility testing results according to the reference method and RAST results for patients treated with piperacillin-tazobactam, cefotaxime, meropenem, and ciprofloxacin as empirical antibiotic treatment[a]

| Microorganism | Outcome | Antibiotic treatment, No. (%) | | | |
|---|---|---|---|---|---|
| | | PTZ | CTX | MER | CIP |
| E. coli | No. of patients receiving treatment | 146 | 230 | 39 | 22 |
| | DD method results | | | | |
| | S | 131 (89.7) | 214 (93.0) | 39 (100.0) | 18 (81.8) |
| | I | 7 (4.8) | 2 (0.9) | 0 | 0 |
| | R | 8 (5.5) | 14 (6.1) | 0 | 4 (18.2) |
| | Readable RAST[b] | 146 (100.0) | 229 (99.6) | 39 (100.0) | 22 (100.0) |
| | RAST S/R[c] | 53 (36.3) | 225 (98.3) | 39 (100.0) | 20 (90.9) |
| | RAST ATU[c] | 93 (63.7) | 4 (1.7) | 0 | 2 (9.1) |
| | CA[d] | 52 (98.1) | 221 (98.2) | 39 (100.0) | 20 (100.0) |
| | VME[d] | 0 | 1 (0.4) | 0 | 0 |
| | ME[d] | 1 (1.9) | 2 (0.9) | 0 | 0 |
| | mE[d] | 0 | 1 (0.4) | 0 | 0 |
| | CA (R)[e] | 0/8 | 13/14 | 0/0 | 2/4 |
| K. pneumoniae complex | No. of patients receiving treatment | 36 | 32 | 15 | 4 |
| | DD method results | | | | |
| | S | 28 (77.8) | 31 (96.9) | 14 (93.3) | 3 |
| | I | 1 (2.8) | 0 | 0 | 0 |
| | R | 7 (19.4) | 1 (3.1) | 1 (6.7)[f] | 1 |
| | Readable RAST | 36 (100.0) | 32 (100.0) | 15 (100.0) | 4 |
| | RAST S/R | 22 (61.1) | 32 (100.0) | 15 (100.0) | 3 |
| | RAST ATU | 14 (38.9) | 0 | 0 | 1 |
| | CA | 22 (100.0) | 32 (100.0) | 14 (93.3) | 3 |
| | VME | 0 | 0 | 1 (6.7)[f] | 0 |
| | ME | 0 | 0 | 0 | 0 |
| | mE | 0 | 0 | 0 | 0 |
| | CA (R) | 0/7 | 1/1 | 0/1 | 0/1 |

[a]Proportions are calculated for $n > 10$. ATU, area of technical uncertainty; CA, categorical agreement; VME, very major error; ME, major error; mE, minor error; PTZ, piperacillin-tazobactam; CTX, cefotaxime; CAZ, ceftazidime; MER, meropenem; CIP, ciprofloxacin.
[b]Readable RAST results (proportion of all analyzed isolates).
[c]RAST-categorized S/R and ATU results (proportion of readable RAST results).
[d]Categorical agreement and errors (proportion of categorized RAST results).
[e]RAST detection of resistance according to the reference method.
[f]This isolate was categorized as R by standard disk diffusion but was further analyzed by BMD and categorized as I (i.e., mE according to the BMD result).

piperacillin-tazobactam-treated resistant isolates. Berinson et al. also investigated the impact of EUCAST RAST on antibiotic treatment decision-making, comparing 51 patients with EUCAST RAST results to a historical control cohort of 54 patients with an AST result 24 h after BC positivity either by Vitek 2 or overnight DD AST. The authors reported that for patients with EUCAST RAST results, the optimal antibiotic treatment was initiated on the same day as culture positivity, while this was not the case for the control group. In addition, the use of EUCAST RAST resulted in six cases of antibiotic therapy escalation in patients with ESBL positivity. Cardot Martin et al. (14) compared patients with EUCAST RAST results to a control group with standard DD testing performed in the afternoons and reported that 100% of the patients with RAST results received effective antibiotic treatment within the same day as BC positivity, compared to 88% of the control group, and that escalation, deescalation, and modification of the antibiotic treatment were induced to a higher extent compared to the control group. The time to effective antibiotic treatment was 8 h, compared to 19 h in the control group, even though not a significant difference (14). Our results confirm that EUCAST RAST can provide clinically relevant information and allow for earlier EAT adjustment than conventional AST methods. Furthermore, our results strengthen this conclusion, as we provide a single time point comparison of AST method results, as opposed to the study by Berinson et al., and add that EUCAST RAST presents reliable and relevant results after only 4 h of incubation, while previous studies included EUCAST RAST readings at several time points (4, 6, and 8 h) (12, 14). Interestingly, Berinson et al. reported only a slightly decreased rate of isolates in the ATU for piperacillin-tazobactam with a longer incubation time and no improvement of the CA (12). This finding is in contrast to those of Cardot-Martin et al.,

**TABLE 4** Susceptibility testing results according to the reference method and RAST results for cefotaxime in patients treated with piperacillin-tazobactam, meropenem, and ciprofloxacin as empirical antibiotic treatment[a]

| Microorganism | Outcome | Antibiotic treatment, No. (%) | | |
| --- | --- | --- | --- | --- |
| | | PTZ | MER | CIP |
| *E. coli* | Total no. of patients | 146 | 39 | 22 |
| | DD method results | | | |
| | S | 129 (88.4) | 26 (66.7) | 20 (90.9) |
| | I | 1 (0.7) | 0 | 0 |
| | R | 16 (11.0) | 13 (33.3) | 2 (9.1) |
| | Readable RAST[b] | 146 (100.0) | 39 (100.0) | 22 (100.0) |
| | RAST S/R[c] | 143 (97.9) | 37 (94.9) | 21 (95.5) |
| | RAST ATU[c] | 3 (2.1) | 2 (5.1) | 1 (4.5) |
| | CA[d] | 141 (98.6) | 37 (100.0) | 21 (100.0) |
| | VME[d] | 1 (0.7) | 0 | 0 |
| | ME[d] | 1 (0.7) | 0 | 0 |
| | mE[d] | 0 | 0 | 0 |
| | CA (R)[e] | 15/16 | 13/13 | 2/2 |
| *K. pneumoniae* complex | Total no. of patients | 36 | 15 | 4 |
| | DD method results | | | |
| | S | 31 (86.1) | 9 (60.0) | 4 |
| | I | 0 | 1 (6.7) | 0 |
| | R | 5 (13.9) | 5 (33.3) | 0 |
| | Readable RAST | 36 (100.0) | 15 (100.0) | 4 |
| | RAST S/R | 36 (100.0) | 13 (86.7) | 4 |
| | RAST ATU | 0 | 2 (13.3) | 0 |
| | CA | 36 (100.0) | 13 (100.0) | 4 |
| | VME | 0 | 0 | 0 |
| | ME | 0 | 0 | 0 |
| | mE | 0 | 0 | 0 |
| | CA (R) | 5/5 | 5/5 | 0/0 |

[a]Proportions are calculated for $n > 10$. ATU, area of technical uncertainty; CA, categorical agreement; VME, very major error; ME, major error; mE, minor error; PTZ, piperacillin-tazobactam; CTX, cefotaxime; CAZ, ceftazidime; MER, meropenem; CIP, ciprofloxacin.
[b]Readable RAST results (proportion of all analyzed isolates).
[c]RAST-categorized S/R and ATU results (proportion of readable RAST results).
[d]Categorical agreement and errors (proportion of categorized RAST results).
[e]RAST detection of resistance according to the reference method.

who reported a decrease in the ATU, from 48% to 25%, comparing 4 and 6 h (14), and Bianco et al., who proposed the recently validated prolonged incubation of 16 to 20 h, which yielded an ATU rate of 4.1% for piperacillin-tazobactam (15). The potential gains in relation to the logistical challenges of a prolonged incubation time for piperacillin-tazobactam should be further analyzed in future studies.

We also evaluated the EUCAST RAST results for cefotaxime for the patients receiving piperacillin-tazobactam, meropenem, or ciprofloxacin as the EAT, as cefotaxime is one

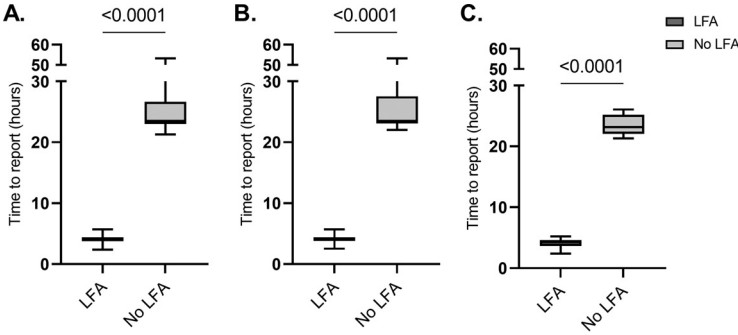

**FIG 2** Time to report of ESBL positivity for ESBL-producing isolates with RAST and positive, reported LFA (LFA) and RAST without reported LFA result (no LFA) for *E. coli* and *K. pneumoniae* isolates ($n = 137$) (A), only *E. coli* isolates ($n = 114$) (B), and only *K. pneumoniae* isolates ($n = 23$) (C).

of our most-prescribed antibiotics and may be a candidate for both escalation and deescalation of the antibiotic treatment. EUCAST RAST correctly categorized all but one of the cefotaxime-resistant isolates and most of the cefotaxime-susceptible isolates. For the patients receiving meropenem as the EAT, all the cefotaxime-resistant isolates were detected, which is indeed important information if the clinician wishes to deescalate meropenem treatment. Furthermore, EUCAST RAST can easily be combined with an LFA to accelerate the reporting of ESBL-producing isolates, which is important both for treatment decisions and to decrease the spread of resistance genes in the hospital.

In our material, there was one *K. pneumoniae* isolate with a BMD result of I for meropenem that was incorrectly categorized as S by EUCAST RAST and R by EUCAST DD, but there were no meropenem-resistant and no carbapenemase-producing isolates. Thus, the present study did not fully evaluate the performance of EUCAST RAST for meropenem. A previous study including 19 meropenem-resistant *E. coli* isolates and 47 meropenem-resistant *K. pneumoniae* isolates reported a CA of 90.2% and 97.2%, respectively, after 6 h of incubation and 95.1% and 95.8% after 8 h of incubation, with 1.4% mE as the only occurring error (17). Another study reported RAST results for 27 meropenem-resistant *E. coli* and *K. pneumoniae* isolates after 4 h of incubation, for which the CA was 99.5% with 1 mE, and only 1% were in the ATU (11). Further verification studies of the EUCAST RAST in settings with high carbapenem resistance could be of interest to further scrutinize the meropenem breakpoints.

In the present study, only *E. coli* and *K. pneumoniae* complex isolates were analyzed, even though EUCAST RAST has been validated for several other bacteria (e.g., *Acinetobacter baumanii*, *Pseudomonas aeruginosa*, and *Staphylococcus aureus*, among others). Our laboratory chose to implement EUCAST RAST only for *E. coli* and *K. pneumoniae* complex isolates, as these bacteria are the most common Gram-negative strains isolated from BCs in the hospitals served by our laboratory. Studies of the diagnostic performance and clinical utility of EUCAST RAST for different species will be useful for further assessment of the method.

A major strength of the present study is the high number of clinical samples with *E. coli* and *K. pneumoniae* complex isolates. In addition, the clinical performance of EUCAST RAST for adjusting the EAT and the possible gains of combining RAST with an LFA for detection of ESBL-producing strains were studied. The limitations of the study were that the information on antibiotic treatment was obtained from the notes recorded in the LIS and not directly from the patients' medical records and that we could not study outcomes in terms of escalation/deescalation, hospital length of stay, health care-associated infections, or mortality. Furthermore, this was a single-center study, and similar multicenter studies analyzing the performance of RAST are warranted. Another limitation is the possibility that a lab technician may have omitted an unreadable RAST plate without recording it, which would overestimate the proportion of readable RAST zones. As it is unusual for an entire plate to be unreadable, we expect this to be less common and thus to have little impact on the results of this study.

**Conclusion.** The present study shows that EUCAST RAST produces reliable and clinically relevant information on resistance for the relevant antibiotics for *E. coli* and *K. pneumoniae* isolates after 4 h of incubation. EUCAST RAST has a high detection rate for cefotaxime resistance and can be combined with a lateral flow assay to accelerate the reporting of ESBL-producing strains. EUCAST RAST is an important tool to shorten the time to appropriate treatment in patients with bacteremia.

## MATERIALS AND METHODS

**Study design.** The present study was conducted at the clinical microbiology laboratory at Karolinska University Hospital (Huddinge, Stockholm, Sweden), which serves one tertiary- and two secondary-care hospitals in Stockholm (Karolinska University Hospital in Huddinge, Stockholm South Hospital, and Södertälje Hospital) and operates 7 days a week during the daytime. EUCAST RAST was implemented for *E. coli* and *K. pneumoniae* isolates in February 2020, in parallel with standard DD testing for BC bottles that signaled positive before 11:00 a.m. all days of the week. AST data from 667 clinical BCs (553 *E. coli* isolates and 114 *K. pneumoniae* isolates) analyzed between February 2020 and March 2021 were compiled. BCs with polymicrobial growth or collected from patients younger than 18 years of age were excluded (Fig. 1). If two strains were isolated from the same patient, the most resistant strain was included.

**Blood cultures.** The BCs retrieved at the laboratory were incubated in the BacT/Alert 3D BC system (bioMérieux, Marcy-l'Étoile, France) at 36°C for 5 days or until signaling positive, according to the standard clinical routine. BacT/Alert FA Plus and BacT/Alert FN Plus BC bottles were used. Positive BCs were Gram stained and

subcultured on agar plates. Species identification was performed using matrix-assisted laser desorption ionization–time of flight mass spectrometry (MALDI-TOF MS; microflex LRF; Bruker Daltonik, Bremen, Germany) with short-term (2- to 4-h) cultures (21).

**RAST.** RAST was performed according to the EUCAST RAST protocol (10). Briefly, 100 to 150 $\mu$L of undiluted broth from positive BC bottles were spread onto 90-mm Mueller-Hinton (MH) agar plates with AST disks for piperacillin-tazobactam (30/6 $\mu$g), cefotaxime (5 $\mu$g), ceftazidime (10 $\mu$g), meropenem (10 $\mu$g), and ciprofloxacin (5 $\mu$g). Both the agar plates and disks were from Oxoid/Thermo Fisher Scientific (Basingstoke, UK). The plates were incubated at 35℃ in ambient air. Zones were read manually after 4 h $\pm$ 5 min, from the front of the plate with the lid off. Zones were not read when there was nonconfluent growth or poorly demarcated zone edges. Interpretation as S or R was performed according to the EUCAST RAST breakpoint tables valid at the time of the study (versions 1.1, 2.0, and 3.0) (22). The RAST zone diameters were recorded continuously in an Excel file developed for the RAST documentation, and the interpretation was reported to clinicians through the LIS.

**Reference method.** The EUCAST standard DD method was performed in parallel (i.e., from subcultured colonies and zone readings after 16 to 20 h of incubation) with media and disks from Oxoid/Thermo Fisher Scientific. The results from the standard DD method were used as the reference (23). Categorization was performed according to the EUCAST breakpoint table for standard DD testing valid at the time of the study (versions 10.0 and 11.0) (24). The EUCAST DD results were recorded by the laboratory staff in the LIS. Selected isolates were further analyzed by broth microdilution (BMD) using the Sensititre system (Thermo Fisher Scientific, MA, USA) according to the manufacturer's instructions.

**ESBL diagnostics.** In April 2020, a qualitative LFA for the detection of ESBL enzymes belonging to CTX-M groups 1, 2, 8, 9, and 25 (NG-Test CTX-M/CTX-M Multi; NG Biotech) was introduced in the laboratory. The test was performed in parallel to the phenotypic confirmatory ESBL double disk synergy test (DDST) for isolates where RAST had detected either cefotaxime or ceftazidime resistance. Bacterial colonies were dissolved in the extraction buffer from the NG-Test, of which 100 $\mu$L was transferred to a test strip. The strip was read after 10 min, with a positive test result identified by the appearance of a colored test line. The DDST was performed by inoculating MH agar plates with a suspension of the bacteria, dissolved in saline to the density of a 0.5 McFarland turbidity standard, whereupon cefotaxime (5 $\mu$g) and ceftazidime (10 $\mu$g) disks were applied 2 to 3 cm apart from an amoxicillin-clavulanic acid (30 $\mu$g) disk. The test was positive if the inhibition zone was enhanced around one of the cephalosporins or if there was a "keyhole" in the direction of the amoxicillin-clavulanic acid disk, after 20 h of incubation at 37℃ (25). The disk and agar manufacturers were those specified above.

**Data collection.** EUCAST RAST zone diameters and interpretations were collected from a RAST Excel file, in which the data were recorded at the time of the reading. The standard DD AST results were collected from the LIS. Information on antibiotic treatment at the time of BC positivity is routinely recorded in the LIS by the clinical microbiologist when reporting the Gram stain result from the positive BC to the treating health care personnel. Data regarding antibiotic treatment, ESBL positivity, time to ESBL report in the LIS from the first report on BC positivity, and patient gender and age were collected from the LIS.

**Evaluation of EUCAST RAST.** The number of readable RAST results was compared to the number of isolates analyzed. The number of categorized S/R results obtained using RAST was compared to the number of readable RAST results. RAST categorical errors were calculated as proportions of the total number of zones categorized as S or R, using the standard DD results as a reference. Errors were defined as very major errors (VMEs; S according to RAST and R according to standard DD), major errors (MEs; R according to RAST and S according to standard DD), or minor errors (mEs; S/R according to RAST and I according to standard DD) (8).

**RAST impact on antibiotic treatment.** Antibiotic treatment at the time of verbal information transfer about BC positivity was used as a proxy for EAT. For patients receiving piperacillin-tazobactam, cefotaxime (or ceftriaxone), meropenem, or ciprofloxacin as the EAT, the final standard DD result and the RAST result were compared to assess the ability of RAST to detect resistance to the EAT. These antibiotics are the most commonly prescribed antibiotic agents in Gram-negative infections and often used as the EAT in Sweden. Furthermore, the RAST result for cefotaxime was assessed for patients receiving piperacillin-tazobactam, meropenem, and ciprofloxacin to evaluate whether RAST could provide valuable information for treatment escalation or deescalation.

**Time to ESBL report in the LIS.** To evaluate the effect on the time to ESBL report in the LIS by combining RAST with an LFA, RAST results from an extended time period between February 2020 and April 2022 were collected. For ESBL-positive isolates where RAST had detected either cefotaxime or ceftazidime resistance, the time to ESBL being reported in the LIS was measured from the time of the Gram stain result. Patients who had a positive LFA result reported in the LIS were compared to patients who did not. From here on, the two groups are referred to as "RAST with positive, reported LFA" and "RAST without reported LFA." The time to report was described as the median and interquartile range (IQR) and compared between the groups using the Mann-Whitney U test due to not normally distributed data. *P* values of <0.05 were considered statistically significant.

All calculations were performed using Microsoft Excel 2021 version 22.02.

**Quality control.** Quality control (QC) according to the EUCAST recommendations was performed daily to control the media and disks used (23). Adherence to the method protocol was ascertained by analysis of spiked BC bottles when introducing new personnel to the RAST method.

**Ethical considerations.** Due to the observational and noninterventional nature of the study, no ethical approval was required. The study did not affect patient treatment or outcome. Laboratory data on BC results, EAT, gender, and age were pseudonymized and analyzed only on a group level.

## ACKNOWLEDGMENTS

We thank the laboratory staff at the clinical microbiology laboratory at Karolinska University Hospital (Huddinge, Stockholm).

A. Ekwall-Larson received funding from Karolinska Institutet. We declare no conflicts of interest.

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
