## [Reviewer comments · Microbiology Spectrum]

Microbiology Spectrum

The analytical performance and potential clinical utility of EUCAST RAST in blood cultures after four hours of incubation

Anna Ekwall-Larson, Inga Fröding, Berivan Mert, Anna Åkerlund, and Volkan Özenci

Corresponding Author(s): Volkan Özenci, Karolinska Institutet

Review Timeline:

Submission Date:	December 6, 2022
Editorial Decision:	December 27, 2022
Revision Received:	January 19, 2023
Accepted:	January 29, 2023

Editor: Karen Carroll

Reviewer(s): Disclosure of reviewer identity is with reference to reviewer comments included in decision letter(s). The following individuals involved in review of your submission have agreed to reveal their identity: Michael Joseph Satlin (Reviewer #4)

Transaction Report:

DOI: <https://doi.org/10.1128/spectrum.05001-22>

December 27, 2022

Prof. Volkan Özenci
Karolinska Institutet
Microbiology
Division of Clinical Microbiology F 72, Karolinska Institutet, Karolinska University Hospital, Huddinge, SE 141 86 Stockholm,
Sweden
Stockholm
Sweden

Re: Spectrum05001-22 (The analytical and clinical performance of EUCAST RAST in blood cultures)

Dear Prof. Volkan Özenci:

Thank you for submitting your manuscript to Microbiology Spectrum. Your paper has been reviewed by three experts on susceptibility testing and all believe that the paper can move forward to publication after revision. I concur with their recommendations. When submitting the revised version of your paper, please provide (1) point-by-point responses to the issues raised by the reviewers as file type "Response to Reviewers," not in your cover letter, and (2) a PDF file that indicates the changes from the original submission (by highlighting or underlining the changes) as file type "Marked Up Manuscript - For Review Only". Please use this link to submit your revised manuscript - we strongly recommend that you submit your paper within the next 60 days or reach out to me. Detailed instructions on submitting your revised paper are below.

Link Not Available

Sincerely,

Karen Carroll

Journals Department
Reviewer comments:

Reviewer #1 (Comments for the Author):

This is a retrospective single center evaluation, including analytical performance and clinical utility, of EUCAST RAST guidelines. Evaluation was performed with *E. coli* and *Klebsiella pneumoniae* complex isolates once identified in blood cultures flagged positives an identified by MALDI TOF. The article is interesting but need to clarify some issues.

General comment

- In the clinical evaluation, only appropriateness of empiric antibiotic therapy was analyzed but no other issues. This evaluation

was just theoretical as potential valuable information for treatment escalation or de-escalation but not through clinical chart review. These are limitations of the study, and should be mentioned as well the single center evaluation. The title should modify accordingly. A suggestion:

"The analytical performance and potential clinical utility of EUCAST RAST in blood cultures"

- Why do the authors did not include in the study a lateral flow for carbapenemases? They had the meropenem results for the implementation.

Specific comments

- Lines 161. Why lateral flow was not performed directly from positive blood culture once the identification was performed with MaldiToF?

- Line 223-224. Although in M&M section clarify that these isolates were just flagged. positive until 11 pm.

Reviewer #2 (Comments for the Author):

The authors present a study on the performance of the EUCAST RAST (rapid AST from blood cultures). Additionally, they analyze the clinically relevant outcomes such as treatment appropriateness.

Notably, the study explicitly investigates the EUCAST RAST after four hours of incubation. This is an interesting aspect of the study as this time point is seen as critical for the reading of disk diffusion. While reading after 6 h and especially 8 h according to EUCAST RAST breakpoints is often regarded as useful, reading after 4 h has been frequently described as practically not feasible due to insufficient growth. Therefore, this study significantly contributes to the knowledge and is of practical relevance.

The methods are technically sound and the results are well described and discussed.

Abstract: "S/R categorized RAST-results were obtained for 83.1 % 38 (2194/2641) and 87.5 % (488/558) for E. coli and K. pneumoniae complex respectively".

What was the reason for not-obtaining S/R categorization. ATU? Please the reason in the abstract. This is important because frequent ATU results seem to considerably limit the application of EUCAST RAST worldwide.

Abstract: "... and almost all cefotaxime resistant isolates that were treated with cefotaxime (13/14 and 1/1 respectively)." It is unclear to what "13/14 and 1/1 respectively" belong. Please rephrase.

Lines 148-149: "The EUCAST standard DD was used as reference (15) and performed in parallel using the same media and disks as described above".

Please state whether the same plates as for RAST were further incubated for reading after 16-20 h or other plates were inoculated for the standard disk diffusion?

Lines 233-235: The highest rates of results in the ATU were seen in piperacillin-tazobactam in both E. coli and K. pneumoniae (62.8 % and 33.9 % respectively)"

This is a very important data from the study - please include it into the abstract. In many areas, piperacillin-tazobactam is the most commonly used antibiotics in sepsis. High number of ATUs with pip/tazo considerably limits the application of the method.

In the discussion section, please explain the choice of bacterial species included into the study. Why Acinetobacter, enterococci, pneumococci (EUCAST RAST breakpoints are also available for 4 h incubation time) were excluded?

Reviewer #4 (Comments for the Author):

This is a real-world evaluation of the implementation of a rapid antimicrobial susceptibility test from EUCAST (RAST) applied directly to positive blood culture bottles for E. coli and Klebsiella pneumoniae at a single microbiology laboratory in Sweden. 530 blood cultures positive for E. coli and 112 positive for K. pneumoniae were included. RAST was read after 4 hours of incubation for five antibiotics and results were compared to EUCAST reference disk diffusion. They also performed a rapid CTX-M lateral flow assay if RAST demonstrated cefotaxime resistance. They found {greater than or equal to}89% of the results were interpretable as S or R at 4 hours for cefotaxime, ceftazidime, meropenem, and ciprofloxacin, but only 37% of the results were interpretable for pip-tazo. Diagnostic performance of interpretable results was generally high. RAST only detected resistance to the empiric antibiotic that was used in 15/26 E. coli and 1/10 K. pneumoniae. The ESBL LFA provided results much earlier than the reference method for ESBL production.

I have the following comments for the authors to respond to:

Major comments:

1) The RAST method seems unreliable (at least at 4-hr read) for pip-tazo. There were a high proportion of ATU results with pip-tazo and of the 15 patients treated empirically with PTZ for PTZ-resistant isolates, PTZ resistance was never detected by RAST. In contrast, performance for cefotaxime was much better. The poor results with pip-tazo should be noted in the Abstract, as well

as the antibiotics evaluated in the study.

2) Table 1: The denominators do not appear correct for the calculation of %VME and %ME. For example, for VME for CTX/E. coli, the denominator should be no greater than 53 (the # of resistant isolates). I recommend recalculating these %s with the appropriate denominators and reporting the denominators (e.g., denominator for VME the # of isolates resistant by reference method that had RAST S or R)

3) It is important to also report the diagnostic accuracy of the ESBL LFA test compared to the reference ESBL double disk synergy test.

4) There are many published studies of the accuracy and clinical implementation of RAST that are not referenced here (Cardot Martin et al. Infect Dis Now 2022; Cherkaoui et al. JCM 2022; Bianco et al. Antibiotics (Basel) 2022, among many others). I think it is important for the authors to place their results into context of these other studies in the Discussion and highlight what unique aspects can be learned from this study.

Minor comments:

1) EUCAST RAST can be performed at multiple times (4, 6, 8, and 16-20 hours). This study evaluated a 4-hour RAST. I think the use of 4-hour reads should be in the Title so that a reader can clearly identify this and put the results into context.

2) It is unclear why blood cultures from children were excluded from this analysis

3) Please clarify whether the reference disk diffusion method was performed on isolated colonies or on the positive blood culture broth.

4) For Table 2: do the authors mean "deceased" instead of "diseased"?

5) The rationale for only including patients treated empirically with pip-tazo, meropenem, and ciprofloxacin as empiric antibiotic therapy (and not the entire dataset), should be more clearly explained. I assume it is because these are patients with opportunities for antibiotic de-escalation, but this is not well explained.

Staff Comments:

Preparing Revision Guidelines

Please return the manuscript within 60 days; if you cannot complete the modification within this time period, please contact me. If you do not wish to modify the manuscript and prefer to submit it to another journal, please notify me of your decision immediately so that the manuscript may be formally withdrawn from consideration by Microbiology Spectrum.

Response to reviewers

Reviewer #1 (Comments for the Author):

This is a retrospective single center evaluation, including analytical performance and clinical utility, of EUCAST RAST guidelines. Evaluation was performed with *E. coli* and *Klebsiella pneumoniae* complex isolates once identified in blood cultures flagged positives and identified by MALDI TOF. The article is interesting but need to clarify some issues.

General comment

1. In the clinical evaluation, only appropriateness of empiric antibiotic therapy was analyzed but no other issues. This evaluation was just theoretical as potential valuable information for treatment escalation or de-escalation but not through clinical chart review. These are limitations of the study, and should be mentioned as well the single center evaluation. The title should modify accordingly. A suggestion:
"The analytical performance and potential clinical utility of EUCAST RAST in blood cultures"

We thank the reviewer for this comment on the external validity of the study. On lines 386-389 we describe the limitations of not studying patient records for actual antibiotic decision-making and evaluation of clinical outcome. We have now added the single-center study design as a limitation as the reviewer suggested. We also adjusted the title in accordance with the reviewer's suggestion and added the word "potential" in the Introduction as well.

Title: "The analytical performance and potential clinical utility of EUCAST RAST in blood cultures after four hours of incubation"

Lines 104-106: "To evaluate the potential clinical usefulness of RAST in antibiotic treatment decision-making, the cefotaxime RAST results for patients treated with piperacillin-tazobactam, meropenem and ciprofloxacin were analyzed."

Lines 388-390: "Furthermore, this was a single center study and similar multi-center studies analyzing the performance of RAST are warranted."

2. Why do the authors did not include in the study a lateral flow for carbapenemases?
They had the meropenem results for the implementation.

*We agree with the reviewer that the meropenem RAST result could be used for a lateral flow carbapenemase test. However, the test was not included in the study design since the prevalence of carbapenemase producing *E. coli* and *K. pneumoniae* is extremely low in our center. Indeed, there was no meropenem R isolate by RAST in the whole study population.*

Specific comments

1. Lines 161. Why lateral flow was not performed directly from positive blood culture once the identification was performed with Malditof?

The rate of ESBL positivity in gram negative bacteria isolated from blood cultures in our clinical microbiology laboratory is around 10-15 %. Performing the lateral flow for E. coli and K. pneumoniae directly from blood cultures would result in many negative results as well as extra work for the laboratory staff, in addition to the cost of the lateral flow assay itself. Therefore, the workflow includes RAST that screens for cefotaxime and ceftazidime resistance in order to reduce the number of lateral flow assays performed.

2. Line 223-224. Although in M&M section clarify that these isolates were just flagged. positive until 11 pm.

The EUCAST RAST method was performed for blood cultures that signaled positive before 11:00 a.m. all days of the week. The reason for this time breakpoint is that the clinical utility of the RAST result is considered much lower if received late in the afternoon after the clinician's working day is over and when there are only clinicians on call left at the hospital. We do not expect the microorganisms in the blood culture bottles that signal positive after 11:00 a.m. to be different in terms of resistance pattern or to be affecting the study's results.

As the reviewer suggested the following sentence was included in the results section of the revised version:

Line 224-225 "Blood cultures with gram-negative bacteria that signaled positive before 11:00 a.m. were included in the study."

Reviewer #2 (Comments for the Author):

The authors present a study on the performance of the EUCAST RAST (rapid AST from blood cultures). Additionally, they analyze the clinically relevant outcomes such as treatment appropriateness.

Notably, the study explicitly investigates the EUCAST RAST after four hours of incubation. This is an interesting aspect of the study as this time point is seen as critical for the reading of disk diffusion. While reading after 6 h and especially 8 h according to EUCAST RAST breakpoints is often regarded as useful, reading after 4 h has been frequently described as practically not feasible due to insufficient growth. Therefore, this study significantly contributes to the knowledge and is of practical relevance.

The methods are technically sound and the results are well described and discussed.

1. Abstract: "S/R categorized RAST-results were obtained for 83.1 % (2194/2641) and 87.5 % (488/558) for E. coli and K. pneumoniae complex respectively".

What was the reason for not-obtaining S/R categorization. ATU? Please the reason in the abstract. This is important because frequent ATU results seem to considerably limit the application of EUCAST RAST worldwide.

As the reviewer proposes the reason for not obtaining a S/R categorization for 16.9 % of the E. coli and 12.5 % of the K. pneumoniae complex is that the zone diameters were in the ATU.

We agree with the reviewer that frequent ATU results is the main limitation to the EUCAST RAST method especially for piperacillin-tazobactam. Even so, we find it difficult to find the room for a description of the ATU in the abstract. The first line of the result section in the abstract describes the readable RAST zones, i.e. how many zones that were included for analysis. In the next sentence we describe the number and percentage of categorized isolates in relation to the included. The corresponding percentage to this is “not categorized”, i.e. ATU. The background to the ATU is presented in the introduction (lines 88-89). We suggest that the ATU will remain explained in the main text.

2. Abstract: "... and almost all cefotaxime resistant isolates that were treated with cefotaxime (13/14 and 1/1 respectively)."

It is unclear to what "13/14 and 1/1 respectively" belong. Please rephrase.

We thank the reviewer for this comment and have rephrased the sentence in the abstract.

*Lines 40-43: “RAST detected 15/26 and 1/10 of the *E. coli* and *K. pneumoniae* complex that were resistant to the EAT. For patients treated with cefotaxime, RAST detected 13/14 cefotaxime-resistant *E. coli* and 1/1 cefotaxime-resistant *K. pneumoniae* complex.”*

3. Lines 148-149: "The EUCAST standard DD was used as reference (15) and performed in parallel using the same media and disks as described above".

Please state whether the same plates as for RAST were further incubated for reading after 16-20 h or other plates were inoculated for the standard disk diffusion?

We thank the reviewer for this comment, as the sentence could be interpreted as we later used the RAST plates for standard DD, which was not the case. The intention of the sentence was that the same manufacturer and type of media and disks were used for both EUCAST RAST and standard DD.

As we have written in the manuscript, the EUCAST standard DD was performed in parallel with EUCAST RAST. Thus, a parallel set of plates were incubated and read after 16-20 hours, and we did not re-incubate the EUCAST RAST plates.

We have clarified this in the revised manuscript, and also that the MH agar used for EUCAST RAST was from Oxoid/Thermo Fisher Scientific, Basingstoke, UK.

Lines 148-150: “The EUCAST standard DD was performed in parallel (i.e. from subcultured colonies and zone readings after 16-20 hours of incubation) with media and disks from Oxoid/Thermo Fisher Scientific, Basingstoke, UK. The results from standard DD were used as reference (20).”

Lines 137-138: “Both agar plates and disks were from Oxoid/Thermo Fisher Scientific, Basingstoke, UK.”

4. Lines 233-235: The highest rates of results in the ATU were seen in piperacillin-tazobactam in both *E. coli* and *K. pneumoniae* (62.8 % and 33.9 % respectively)" This is a very important data from the study - please include it into the abstract. In many areas, piperacillin-tazobactam is the most commonly used antibiotics in sepsis. High number of ATUs with pip/tazo considerably limits the application of the method.

We agree with the reviewer that the low rate of categorized S/R results for piperacillin-tazobactam is an important result that should be presented in the abstract. We have included a description of this result in the revised version.

Line 38-39: "S/R categorization for piperacillin-tazobactam was poor (37.2 % and 66.1 % respectively)."

5. In the discussion section, please explain the choice of bacterial species included into the study. Why *Acinetobacter*, enterococci, pneumococci (EUCAST RAST breakpoints are also available for 4 h incubation time) were excluded?

*When introducing the EUCAST RAST method in our clinical microbiology laboratory, the lab chose to only perform RAST on gram-negative isolates and to only report results for *E. coli* and *K. pneumoniae* complex. The reason for this was that these isolates are the most common gram-negatives isolated from blood cultures in our lab. We have included the following explanation in the Discussion section:*

*Lines 374-380: "In the present study only *E. coli* and *K. pneumoniae* complex were analyzed, even though EUCAST RAST has been validated for several other bacteria (e.g., *Acinetobacter baumannii*, *Pseudomonas aeruginosa*, *Staphylococcus aureus* among others). The reason was that our laboratory chose to implement EUCAST RAST only for *E. coli* and *K. pneumoniae* complex, as these bacteria are the most common gram-negatives isolated from BCs in the hospitals served by our laboratory. Studies of diagnostic performance and clinical utility of EUCAST RAST for different species will be useful for further assessment of the method."*

Reviewer #4 (Comments for the Author):

This is a real-world evaluation of the implementation of a rapid antimicrobial susceptibility test from EUCAST (RAST) applied directly to positive blood culture bottles for *E. coli* and *Klebsiella pneumoniae* at a single microbiology laboratory in Sweden. 530 blood cultures positive for *E. coli* and 112 positive for *K. pneumoniae* were included. RAST was read after 4 hours of incubation for five antibiotics and results were compared to EUCAST reference disk diffusion. They also performed a rapid CTX-M lateral flow assay if RAST demonstrated cefotaxime resistance. They found {greater than or equal to}89% of the results were interpretable as S or R at 4 hours for cefotaxime, ceftazidime, meropenem, and ciprofloxacin,

but only 37% of the results were interpretable for pip-tazo. Diagnostic performance of interpretable results was generally high. RAST only detected resistance to the empiric antibiotic that was used in 15/26 *E. coli* and 1/10 *K. pneumoniae*. The ESBL LFA provided results much earlier than the reference method for ESBL production.

I have the following comments for the authors to respond to:

Major comments:

1. The RAST method seems unreliable (at least at 4-hr read) for pip-tazo. There were a high proportion of ATU results with pip-tazo and of the 15 patients treated empirically with PTZ for PTZ-resistant isolates, PTZ resistance was never detected by RAST. In contrast, performance for cefotaxime was much better. The poor results with pip-tazo should be noted in the Abstract, as well as the antibiotics evaluated in the study.

We agree with the reviewer that the EUCAST RAST method's performance with piperacillin-tazobactam is important and should be presented in the abstract. We also agree that specification of the antibiotics tested should be included in the abstract. We have included both in the revised version of the manuscript.

Line 38-39: S/R categorization for piperacillin-tazobactam was poor (37.2 % and 66.1 % respectively)."

Lines 31-35: "The rate of categorized RAST results and the categorical agreement (CA) of RAST with standard EUCAST 16-20 h disk diffusion (DD) for piperacillin-tazobactam, cefotaxime, ceftazidime, meropenem and ciprofloxacin were analyzed, as well as the utility of RAST for adjusting empiric antibiotic therapy (EAT) and combining RAST with a lateral flow assay (LFA) for ESBL detection.

2. Table 1: The denominators do not appear correct for the calculation of %VME and %ME. For example, for VME for CTX/*E. coli*, the denominator should be no greater than 53 (the # of resistant isolates). I recommend recalculating these %s with the appropriate denominators and reporting the denominators (e.g., denominator for VME the # of isolates resistant by reference method that had RAST S or R)

Although we appreciate that the recommendations for acceptance testing of new AST-products (FDA regulation, the previous ISO-standard 20776-2:2007) for approval of a new method were that error calculations should be performed as the reviewer points out, this has not been generally implemented in previous publications. Note however, that in the updated ISO standard for AST method evaluations, ISO-20776-2:2021, error calculations have been replaced by sensitivity and specificity calculations for qualitative AST methods. Our study, which is not an AST acceptance trial, but an evaluation of the performance in routine laboratory work, is based on routine samples and does not include the sufficient number of challenge isolates and resistant isolates for an AST acceptance trial. Error rates are always highly influenced of the number and type/degree of resistance of the included isolates,

regardless of which method of calculation is used. In addition, the added complication of ATU in evaluations of the EUCAST RAST method further complicates the issue.

In order to facilitate comparisons with previous studies, we have chosen to use the same denominator as in the two original EUCAST RAST publications (1,2) where errors were calculated on the total number of zones interpreted as S or R. Other studies have also chosen this way of presenting errors (3,4). We have stated the number of isolates with each susceptibility result in Tables 1, 3 and 4 to provide the reader with sufficient data to perform additional calculations if desired. The error calculation is explained in M&M lines 184-185 and in the table footnotes.

3. It is important to also report the diagnostic accuracy of the ESBL LFA test compared to the reference ESBL double disk synergy test.

The goal of this part of the study was to analyze the time-gain using an ESBL LFA test performed after EUCAST RAST. The performance of the ESBL LFA method was not analyzed in the present study. The sensitivity and specificity cannot be analyzed since the LFA result were not always recorded in the LIS and/or not reported to the clinics. Only positive test results that were reported were taken into account in the analysis to assess the turnaround time for ESBL detection.

4. There are many published studies of the accuracy and clinical implementation of RAST that are not referenced here (Cardot Martin et al. Infect Dis Now 2022; Cherkaoui et al. JCM 2022; Bianco et al. Antibiotics (Basel) 2022, among many others). I think it is important for the authors to place their results into context of these other studies in the Discussion and highlight what unique aspects can be learned from this study.

We thank the reviewer for this relevant comment. Three references were accidentally removed at some point during editing of the manuscript (Soo et al. 2020, Jasuja et al. 2020 and Martins et al. 2020). These references are now re-included. As the reviewer points out, several relevant studies were unfortunately not included in the Discussion and we agree that our results should be discussed in relation to these studies. We have now revised the Discussion and added several results from other research work to assess our own results more thoroughly.

We have not included Cherkaoui et al. as it is stated in the article that another dilution of the blood for incubation was used in the automated method than in the standard EUCAST RAST protocol. We think that this methodological difference limits the relevance of comparison of our results with that study.

Line 97: Addition of references Martins et al. 2020, Cardot Martin et al. 2022, Bianco et al. 2022, Valentin et al. 2021, Shan et al. 2022.

Line 98: Addition of reference Cardot Martin et al. 2022.

Line 300: Addition of references Martins et al. 2020, Cardot Martin et al. 2022, Bianco et al. 2022, Soo et al. 2020, Jasuja et al. 2020.

Line 302- 303: “Previous studies have also reported high rate of errors for piperacillin-tazobactam”. Addition of references Soo et al. 2020, Jasuja et al. 2020, Martins et al. 2020

Lines 316-321: “This is in line with a previous study in our laboratory on short-term incubation of EUCAST disk diffusion, ROSCO ESBL and carbapenemase detection kits in which high performance for detection of cefotaxime resistance was reported (25), and Cortazzo et al. who reported a CA of 100 % for cefotaxime and E. coli and K. pneumoniae after four, six and eight hours of incubation including detection of all 48 ESBL producing isolates (11).”

Lines 332-338:” Cardot Martin et al. compared patients with EUCAST RAST results to a control group with standard DD performed in the afternoons and reported that 100 % of the patients with RAST results received effective antibiotic treatment within the same day as BC positivity compared to 88 % of the control group, and induced both escalation, de-escalation and modification of the antibiotic treatment to a higher extent compared to the control group. Time to effective antibiotic treatment was 8 h compared to 19 h in the control group, even though not a significant difference (14).”

Lines 340-344: “Furthermore, our results strengthen this conclusion as it is a single time point comparison of AST method results as opposed to the study of Berinson et al. and adds that EUCAST RAST presents reliable and relevant results after only four hours of incubation, while previous studies included EUCAST RAST readings at several time points (four, six and eight hours) (12,14).” (Addition of reference Cardot Martin et al. 2022).

Lines 346-351: “This contrasts to Cardot-Martin et al. who reported a decrease in ATU from 48 % to 25 % comparing four and six hours (14) and Bianco et al. who propose the recently validated prolonged incubation of 16-20 h which yielded an ATU rate of 4.1 % for piperacillin-tazobactam (15). The potential gains in relation to logistical challenges of a prolonged incubation time for piperacillin-tazobactam should be further analyzed in future studies.”

Lines 365-368: A previous study including 19 meropenem resistant *E. coli* and 47 meropenem resistant *K. pneumoniae* reported a CA of 90.2 % and 97.2 % respectively after 6 hours of incubation and 95.1 % and 95.8 % after 8 hours of incubation, with 1.4 % mE as the only occurring error (17).

Minor comments:

1. EUCAST RAST can be performed at multiple times (4, 6, 8, and 16-20 hours). This study evaluated a 4-hour RAST. I think the use of 4-hour reads should be in the Title so that a reader can clearly identify this and put the results into context.

We agree with the reviewer and have adjusted the title accordingly.

Title: “The analytical performance and potential clinical utility of EUCAST RAST in blood cultures after four hours of incubation.”

2. It is unclear why blood cultures from children were excluded from this analysis

We wanted to keep the study material as homogeneous as possible and therefore we chose to exclude children.

1. Please clarify whether the reference disk diffusion method was performed on isolated colonies or on the positive blood culture broth.

The reference disk diffusion method was performed on isolated colonies after subculturing of positive BC bottles.

2. For Table 2: do the authors mean "deceased" instead of "diseased"?

We mean deceased and thank the reviewer for this correction.

3. The rationale for only including patients treated empirically with pip-tazo, meropenem, and ciprofloxacin as empiric antibiotic therapy (and not the entire dataset), should be more clearly explained. I assume it is because these are patients with opportunities for antibiotic de-escalation, but this is not well explained.

We included patients treated with cefotaxime, piperacillin-tazobactam, meropenem and ciprofloxacin for analysis of appropriateness and possible correction of inadequate antibiotic treatment, as these the most used antibiotic agents for gram negative infections and therefore often used as EAT in Sweden. In the next step, we analyzed the EUCAST RAST method's ability to correctly classify cefotaxime resistance for patients receiving piperacillin-tazobactam, meropenem and ciprofloxacin, as cefotaxime is the most commonly used agent for gram-negative bacteremia in Sweden and is an option for both escalation (if the patients received ciprofloxacin as EAT) and de-escalation (if the patient received meropenem or piperacillin-tazobactam). We have included the rationale of analyzing EAT with cefotaxime, piperacillin-tazobactam, meropenem and ciprofloxacin in the revised manuscript, see below. The rationale of the second part of the analysis is already described in M&M (lines 195-197) and Discussion (lines 353-356).

Lines 193-195: These antibiotics are the most commonly prescribed antibiotic agents in gram-negative infections and often used as EAT in Sweden.

References

1. Jonasson E, Matuschek E, Kahlmeter G. The EUCAST rapid disc diffusion method for antimicrobial susceptibility testing directly from positive blood culture bottles. *Journal of Antimicrobial Chemotherapy*. 2020 Apr 1;75(4):968–78.
2. Åkerlund A, Jonasson E, Matuschek E, Serrander L, Sundqvist M, Kahlmeter G, et al. EUCAST rapid antimicrobial susceptibility testing (RAST) in blood cultures: Validation in 55 european laboratories. *Journal of Antimicrobial Chemotherapy*. 2020 Nov 1;75(11):3230–8.
3. Cortazzo V, Giordano L, D’Inzeo T, Fiori B, Brigante G, Luzzaro F, et al. EUCAST rapid antimicrobial susceptibility testing of blood cultures positive for *Escherichia coli* or *Klebsiella pneumoniae*: Experience of three laboratories in Italy. Vol. 76, *Journal of Antimicrobial Chemotherapy*. Oxford University Press; 2021. p. 1110–2.
4. Berinson B, Olearo F, Both A, Brossmann N, Christner M, Aepfelbacher M, et al. EUCAST rapid antimicrobial susceptibility testing (RAST): Analytical performance and impact on patient management. *Journal of Antimicrobial Chemotherapy*. 2021 May 1;76(5):1332–8.

January 29, 2023

Prof. Volkan Özenci
Karolinska Institutet
Microbiology
Division of Clinical Microbiology F 72, Karolinska Institutet, Karolinska University Hospital, Huddinge, SE 141 86 Stockholm,
Sweden
Stockholm
Sweden

Re: Spectrum05001-22R1 (The analytical performance and potential clinical utility of EUCAST RAST in blood cultures after four hours of incubation)

Dear Prof. Volkan Özenci:

Your manuscript has been accepted, and I am forwarding it to the ASM Journals Department for publication. You will be notified when your proofs are ready to be viewed.

Sincerely,

Karen Carroll
Editor, Microbiology Spectrum
